# RRO: LLM Agent Optimization Through Rising Reward Trajectories

**Zilong Wang**[1] **Jingfeng Yang**[2] **Sreyashi Nag**[2] **Samarth Varshney**[2]

**Xianfeng Tang**[2] **Haoming Jiang**[2] **Jingbo Shang**[1] **Sheikh Muhammad Sarwar**[2]

[1]University of California, San Diego [2]Amazon

{zlwang, jshang}@ucsd.edu  smsarwar@amazon.com

## Abstract

Large language models (LLMs) have exhibited extraordinary performance in a variety of tasks while it remains challenging for them to solve complex multi-step tasks as agents. In practice, agents are sensitive to the outcome of certain key steps which makes them likely to fail the task because of a subtle mistake in the planning trajectory. Recent approaches resort to calibrating the reasoning process through reinforcement learning. They reward or penalize every reasoning step with process supervision, known as Process Reward Models (PRMs). However, PRMs are difficult and costly to scale up with a large number of next action candidates since they require extensive computations to acquire the training data through per-step trajectory exploration. To mitigate this issue, we focus on the *relative reward trend* across successive reasoning steps and propose maintaining an increasing reward in the collected trajectories for process supervision, which we term *Reward Rising Optimization* (RRO). Specifically, we incrementally augment the process supervision until we identify a step exhibiting positive reward differentials, i.e. *rising rewards*, relative to its preceding iteration. This method dynamically expands the search space for the next action candidates, efficiently capturing high-quality data. We provide mathematical groundings and empirical results on the WebShop and InterCode-SQL benchmarks, showing that our proposed RRO method achieves superior performance while requiring much less exploration cost.

## 1 Introduction

Large language models (LLMs) have achieved remarkable advancements in numerous domains, ranging from natural language understanding to code generation (Brown et al., 2020; Kojima et al., 2022; Wei et al., 2022; Chen et al., 2023b; Zhou et al., 2023; Zhong et al., 2024). Their ability to process and generate human-like text has significantly expanded their utility in real-world applications. However, solving complex multistep tasks that require reasoning and decision-making capabilities continue to pose substantial challenges for these models (Lightman et al., 2024; Wang et al., 2024b; Song et al., 2024). This limitation has led researchers to explore novel methodologies to enhance the reasoning abilities of LLMs, particularly in the context of agentic tasks requiring temporal sequential decision-making based on the environment feedback (Song et al., 2024; Xiong et al., 2024).

Recent efforts have introduced the reinforcement learning framework into the agent training process. They leverage Process Reward Models (PRMs) to evaluate and guide every reasoning step (Wang et al., 2024c; Luo et al., 2024; Xiong et al., 2024). PRMs utilize reinforcement signals from each step to encourage effective reasoning trajectories while penalizing suboptimal steps, thereby enhancing problem-solving. Despite their promise, the scalability of PRMs remains a tricky problem. Acquiring training data for these models often relies on the next action exploration for every reasoning step, which is computationally expensive and time-intensive (Wang et al., 2024b; Xiong et al., 2024; Luo et al., 2024). Such requirements

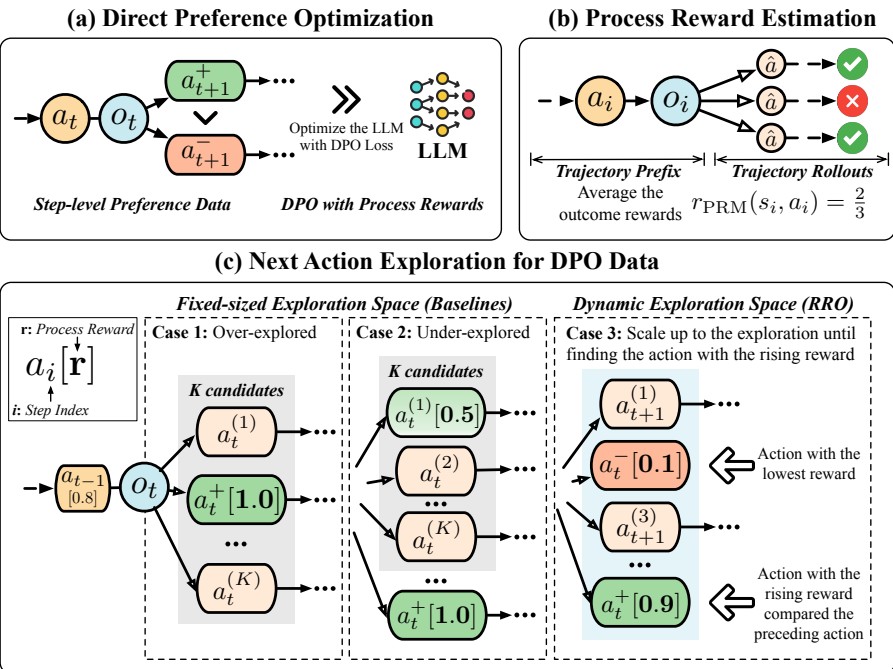

Figure 1: **Reward Rising Optimization:** We dynamically adjust the scope of the next action exploration and prioritize action steps that exhibit a "rising reward trend" compared to their predecessors, avoiding the over-exploration or under-exploration. (a) The illustration of Direct Preference Optimization (Rafailov et al., 2024) where a pair of the preference data is used to optimize the LLM agent. (b) The process reward estimation used in RRO where the average outcome reward of a set of rollouts serves as the process reward of an intermediate step. (c) The comparison of different strategies in the next action exploration stage where our RRO achieves a balance between the computation and data quality.

limit the feasibility of deploying PRMs at scale, particularly in scenarios requiring extensive exploration of decision paths.

In this paper, we address these challenges by re-thinking the process supervision paradigm. Rather than uniformly expanding data collection efforts for every step, we propose a novel approach, *Reward Rising Optimization (RRO)*, that prioritizes actions exhibiting a "rising reward" trend compared to their predecessors, as shown in Figure 1. By focusing on the relative reward tendencies of neighboring reasoning steps, our method dynamically adjusts the scope of candidate action exploration to efficiently identify high-quality data points. This strategy significantly reduces the computational burden in terms of the number of action candidates that need to be explored, while preserving the ability to capture critical decision-making patterns, enabling scalable process supervision for LLMs.

We provide theoretical foundations to support our approach, demonstrating the feasibility of exploring the action candidate with the rising rewards. Additionally, we conduct comprehensive empirical evaluations on automated agent benchmarks, WebShop (Yao et al., 2022) and InterCode-SQL (Yang et al., 2023). The results highlight the efficacy of our method in improving task performance while maintaining computational efficiency. By redefining the data collection paradigm for process supervision, our work paves the way for scalable and effective reinforcement learning frameworks in enhancing the reasoning capabilities of LLMs. We summarize our contribution as follows,

- We propose Reward Rising Optimization (RRO), a scalable process supervision framework that prioritizes reasoning steps with rising reward trends, reducing the burden of extensive data collection.

- The proposed dynamic expansion strategy for sampling next action candidates efficiently identifies and captures high-quality data for training.
- We provide theoretical insights and empirical evidence demonstrating the feasibility of the dynamic expansion strategy and the improved performance and computational efficiency of RRO, on diverse benchmarks, InterCode-SQL and WebShop.

## 2 Preliminary

**Task Formulation** Automated agents play a crucial role in real-world applications. The overall paradigm can be formulated as a partially observable Markov decision process (POMDP) defined by the elements $(\mathcal{U}, \mathcal{S}, \mathcal{A}, \mathcal{O}, \mathcal{T}, \mathcal{R})$. Here, we denote the instruction space as $\mathcal{U}$, the state space as $\mathcal{S}$, the action space as $\mathcal{A}$, the observation space as $\mathcal{O}$, the transition function as $\mathcal{T} : \mathcal{S} \times \mathcal{A} \rightarrow \mathcal{S}$, and the reward function as $\mathcal{R} : \mathcal{S} \times \mathcal{A} \rightarrow [0, 1]$. Since we investigate automated agents powered by LLMs in this paper, $\mathcal{U}, \mathcal{A}, \mathcal{O}$ are represented in the form of natural language. Formally, given a task description $u \in \mathcal{U}$, the LLM-based agent, serving as a policy model $\pi_\theta$, generates the next action $a \in \mathcal{A}$ and observes the execution result $o \in \mathcal{O}$, iteratively. The whole trajectory follows the ReAct style (Yao et al., 2023) and can be formulated as $e = [u, a_1, o_1, ..., a_{n-1}, o_{n-1}, a_n]$, where $n$ is the number of steps, $a_i \sim \pi_\theta(\cdot | u, a_1, o_1, ..., o_{i-1}) \in \mathcal{A}$, and the last action $a_n$ serves as the final outcome from the agent. The interaction loop between the agent and the execution environment repeats until the task completes or exceeds the maximum iteration limit.

**Outcome Supervision** The outcome reward model (ORM) rewards or penalizes the policy model based on the final outcome of a trajectory, which is formulated as $r_{\text{ORM}}(u, e) \in [0, 1]$, where $u$ is the task description and $e$ is the generated trajectory. This outcome supervision evaluates the ultimate success of the agent in accomplishing the specified task rather than focusing on intermediate steps. The normalized reward range $[0, 1]$ provides a standardized measure of task completion quality, with higher values indicating more successful outcomes. Unlike process-level rewards that provide feedback on individual actions, outcome supervision captures the holistic performance of the entire action sequence, enabling the model to learn task completion strategies that may involve different valid approaches to reach the same successful outcome.

**Process Supervision** To further enhance the capability of LLM-based agents, process supervision has been proposed to capture the fine-grained rewards for each action in the trajectory (Lightman et al., 2023; Xiong et al., 2024), also known as the process reward model (PRM). Unlike outcome supervision that evaluates only the final result, process supervision provides feedback on individual actions, enabling more precise guidance throughout the decision-making sequence. Formally, we denote a prefix of trajectory as $e_{1:t} = [u, a_1, o_1, ..., a_t, o_t]$, where $u$ is the task description, $a_i$ represents actions, and $o_i$ represents observations or environment feedback. The process reward model is represented as $r(s_t, a_t) \in [0, 1]$, where $s_t$ and $a_t$ are the state and action at the $t^{\text{th}}$ step. This formulation allows for evaluating the quality of each decision point within the trajectory. Recent work has leveraged Monte Carlo tree search (MCTS) (Wang et al., 2024a;c; Xiong et al., 2024) to build effective step-wise reward models. In this paper, we also adopt the MCTS-based process reward model which is formulated as:

$$r_{\text{PRM}}(s_t, a_t) = \frac{1}{m} \sum_j^m r_{\text{ORM}}(u, e_{1:t} \oplus \hat{e}_{t+1:n}^{(j)})$$

where $\oplus$ represents the concatenation operation and $\hat{e}_{t+1:n}^{(j)}$ denotes one of the sampled rollouts subsequent to the prefix $e_{1:t}$. The approach involves sampling $m$ possible future trajectories from the current state-action pair and calculating the average of their corresponding outcome rewards. This Monte Carlo estimation provides a measure of the expected future return when taking action $a_t$ at state $s_t$.

# 3 Reward Rising Optimization

As introduced in the preliminary (§2), process supervision is promising in improving LLM-based agents. However, how to effectively sample candidates during next action exploration and construct a robust training set for reinforcement learning algorithms remains an open question. In this paper, we focus on the relative reward trend of successive reasoning steps and propose to scale up the exploration until we find a step with *rising rewards* compared to the previous step. The complete pipeline of our proposed Reward Rising Optimization (RRO) contains three stages: (1) supervised fine-tuning (§3.1); (2) reward rising sampling (§3.2); and (3) agent optimization (§3.4).

## 3.1 Supervised Fine-tuning

Similar to previous work (Song et al., 2024; Xiong et al., 2024), we introduce the basic knowledge of the targeted task by first fine-tuning our base model with a supervised dataset to develop basic planning capabilities and adhere to the expected format for each task. We denote this fine-tuned model as $\pi_{\text{SFT}}$. The supervised dataset consists of expert trajectories for the agent tasks, providing demonstrations of successful task completion sequences.

We denote the supervised dataset as $\mathcal{D} = \left\{ (u, e)^{(i)} \right\}_{i=1}^{|\mathcal{D}|}$ where $u$ represents the user instruction or task specification, and $e = [u, a_1, o_1, ..., a_t]$ is the corresponding expert trajectory. The loss function for the supervised fine-tuning stage is formulated as:

$$\mathcal{L}_{\text{SFT}}(\pi_\theta) = -\mathbb{E}_{(u,e)\sim\mathcal{D}} \left[ \log \pi_\theta(e|u) \right] = -\mathbb{E}_{(u,e)\sim\mathcal{D}} \left[ \sum_{t=1}^{n} \log \pi_\theta(a_t|e_{1:t-1}) \right]$$

This formulation decomposes the trajectory modeling problem into a sequence of next-action predictions, where $a_t$ is the action subsequent to the trajectory prefix $e_{1:t-1}$. By maximizing the log likelihood of expert trajectories with this loss function, $\pi_{\text{SFT}}$ learns to imitate the decision-making process demonstrated in the expert data. This supervised learning phase is crucial as it provides the initial policy that will later be refined through preference optimization techniques.

## 3.2 Reward Rising Sampling

After the supervised fine-tuning phase, the supervised fine-tuned model $\pi_{\text{SFT}}$ acquires fundamental planning capabilities for the target agent task and can function effectively as a policy model. The subsequent challenge lies in better aligning this policy with the specific requirements and objectives of the agent task. Before delving into the agent optimization paradigm (§3.4), we introduce in this section our novel Reward Rising Sampling approach, which dynamically scales the exploration process for next action candidates while efficiently capturing the reward rising tendency.

The preference data for process supervision consists of contrastive action pairs $(a_t^+, a_t^-)$ where $a_t^+ \succ a_t^- | e_{1:t-1}$, indicating that action $a_t^+$ is preferred over action $a_t^-$ given the trajectory prefix $e_{1:t-1}$. To generate these contrastive pairs, we sample multiple action candidates using the policy model: $a_t^{(i)} \in \{ \hat{a}_t | \hat{a}_t \sim \pi_{\text{SFT}}(\cdot|e_{1:t-1}) \}$

A straightforward approach to constructing contrastive action pairs would be to select the action candidate with the highest process reward as the chosen sample and the action candidate with the lowest process reward as the rejected sample. However, this naive approach presents a critical trade-off: scaling up the sampling size would increase the probability of generating stronger preferred candidates but would simultaneously introduce substantial computational costs. This raises an important question: what criteria should determine when to stop the sampling process, ensuring that the sampled action candidate is sufficiently strong to serve as a good preferred sample?

Our key insight is to focus on the relative reward tendency between neighboring actions rather than pursuing an absolute maximum. Specifically, we stop scaling up the sampling

---

**Algorithm 1:** Reward Rising Optimization (RRO)

---

**Data:** $u$ is the task instruction; $\pi_\theta$ is the supervised fine-tuned policy model; $e_{1:t}$ is the trajectory prefix from step 1 to $t$; $m$ is the number of samples in Monte-Carlo Tree Search.

1 **Function** ProcessReward($u$, $e_{1:t} = [u, a_1, o_1, ..., a_t, o_t]$, $\pi_\theta$, $m$):
2    $MC \leftarrow [\ ]$
3    **for** $i = 1$ *to* $m$ **do**
4      $\hat{e}_{t+1:n} \sim \pi_\theta(\cdot|e_{1:t})$            ▷ Sample the rollout $\hat{e}_{t+1:n}$ from the policy model $\pi_\theta$
5      $e \leftarrow e_{1:t} \oplus \hat{e}_{t+1:n}$     ▷ Concatenate the trajectory prefix and rollout to have the full trajectory $e$.
6      $MC$.append($r_{ORM}(u, e)$)       ▷ Evaluate the outcome reward based on the full trajectory.
7    **end**
8    $r_{PRM} \leftarrow \frac{1}{m} \sum_{r_{ORM} \in MC} r_{ORM}$      ▷ Estimate the process reward with the average outcome rewards.
9 **return** $r_{PRM}$

10
11 **Function** RRO($u$, $\pi_\theta$, $t$):
12    $e_{1:t-1} = [u, a_1, o_1, ..., a_{t-1}, o_{t-1}] \sim \pi_\theta(u)$     ▷ Sample $t-1$ steps as the trajectory prefix.
13    $r_{PRM,t-1} \leftarrow$ ProcessReward($u$, $e_{1:t-1}$, $\pi_\theta$, $m$)   ▷ Estimate the process reward via Monte-Carlo Sampling.
14    $Cand \leftarrow [\ ]$
15    **repeat**
16      $\hat{a}_t \sim \pi_\theta(\cdot|e_{1:t-1})$           ▷ Explore the candidates for the next action.
17      $e_{1:t} \leftarrow e_{1:t} \oplus \hat{a}_t$
18      $r_{PRM,t} \leftarrow$ ProcessReward($u$, $e_{1:t}$, $\pi_\theta$, $m$)   ▷ Estimate the process reward for the next action candidate.
19      $Cand$.append($(\hat{a}_t, r_{PRM,t})$)
20    **until** $r_{PRM,t} \geq r_{PRM,t-1}$      ▷ Stop the exploration until showing a rising reward compared to $a_{t-1}$.
21    $a_t^+ \leftarrow \arg\max_{r_{PPM,t}} \{\hat{a}_t | (\hat{a}_t, r_{PRM,t}) \in Cand\}$
22    $a_t^- \leftarrow \arg\min_{r_{PPM,t}} \{\hat{a}_t | (\hat{a}_t, r_{PRM,t}) \in Cand\}$
23    $\pi_\theta \leftarrow$ DPO($\pi_\theta$, $(a_t^+, a_t^-)$)      ▷ Update the policy model with the preference pair from the sampling.

---

size when we identify an action candidate with a higher process reward compared to its preceding action, as detailed in Algorithm 1. This approach offers a more efficient exploration strategy while maintaining the quality of preference data.

The algorithm operates as follows: suppose our last action in the trajectory prefix is $a_{t-1}$ with a process reward of $r_{t-1} = r(s_{t-1}, a_{t-1})$, we sample the next action candidate and calculate the corresponding process reward iteratively:

$$a_t^{(i)} \in \{\hat{a}_t | \hat{a}_t \sim \pi_\theta(\cdot|e_{1:t-1})\}; r_t^{(i)} = r(s_t, a_t^{(i)})$$

In this iterative sampling process, we compare the reward of each sampled action with that of its preceding action in the trajectory, meaning we compare $r_{t-1}$ and $r_t^{(i)}$. Once we sample an action $a_t^{(\tau)}$ whose process reward $r_t^{(\tau)}$ is greater than or equal to the preceding action's reward $r_{t-1}$, we terminate the sampling process and designate $a_t^{(\tau)}$ as our preferred sampled action in the preference data. We then select the candidate with the lowest process reward encountered during sampling as the rejected action in the preference data.

This reward rising criterion provides an efficient stopping condition that balances exploration quality with computational efficiency. By identifying actions that improve upon the current reward state, we can construct meaningful preference pairs without exhaustively sampling the action space. This approach is particularly valuable in complex agent environments where the action space is vast and the computation of rewards can be resource-intensive.

### 3.3 Math Grounding

In this section, we provide math grounding on the feasibility of the dynamic sampling strategy based on the process reward setting adopted in RRO. Following previous work,

we define the process reward as:

$$r_{\text{PRM}}(s_t, a_t) = \frac{1}{m} \sum_{j=1}^{m} r_{\text{ORM}}(u, e_{1:t} \oplus \hat{e}_{t+1:n}^{(j)})$$

which represents the expected success probability of the current action. Expanding this further, we express the process reward as:

$$r_{\text{PRM}}(s_t, a_t) = \frac{1}{m} \sum_{j=1}^{m} r_{\text{ORM}}(u, e_{1:t} \oplus \hat{e}_{t+1:n}^{(j)}) = P(1|u, a_1, o_1, \ldots, a_t, o_t) = P(1|e_{1:t}).$$

Using the law of total probability, we rewrite the probability of success as:

$$P(1|e_{1:t}) = \sum_{a_{t+1}} P(1|e_{1:t}, a_{t+1})P(a_{t+1}|e_{1:t}) = \sum_{a_{t+1}} P(1|e_{1:t+1})P(a_{t+1}|e_{1:t}).$$

We now replace the probability terms with their corresponding process rewards:

$$r_{\text{PRM}}(s_t, a_t) = \sum_{a_{t+1}} P(1|e_{1:t+1})P(a_{t+1}|e_{1:t}) = \sum_{a_{t+1}} r_{\text{PRM}}(s_{t+1}, a_{t+1})P(a_{t+1}|e_{1:t})$$

Since $P(a_{t+1}|e_{1:t}) \in [0, 1]$, there must exist at least one sampled action $a_{t+1}^{(\tau)}$ such that its corresponding process reward satisfies:

$$r_{\text{PRM}}(s_{t+1}, a_{t+1}^{(\tau)}) \geq r_{\text{PRM}}(s_t, a_t).$$

This result follows from the fact that the process reward is an expectation over multiple sampled outcomes, and at least one of them must meet or exceed the previous step's reward. Furthermore, this formulation aligns naturally with the Bellman equation, reinforcing the iterative improvement of the expected success probability.

### 3.4 Agent Optimization

We follow the classic Direct Preference Optimization (DPO) training paradigm and continuously train the supervised fine-tuned model $\pi_{\text{SFT}}$ through DPO on the collected preference data. This approach enables us to effectively align the model with human preferences without requiring a separate reward model. The resulting model is denoted as $\pi_{\text{DPO}}$.

The DPO training objective is formulated as:

$$\mathcal{L}_{\text{DPO}}(\pi_\theta) = -\mathbb{E}_{(x, y_w, y_l)} \left[ \log \left( \sigma \left( \beta \log \frac{\pi_\theta(y_w)}{\pi_{\text{ref}}(y_w)} - \beta \log \frac{\pi_\theta(y_l)}{\pi_{\text{ref}}(y_l)} \right) \right) \right]$$

where $(x, y_w, y_l)$ represents a preference data point with input prompt $x$ and a pair of model responses where $y_w \succ y_l$ indicates our sampled preference data shows $y_w$ is preferred over $y_l$. The action $y_w$ is considered superior to $y_l$ under the policy model $\pi_\theta$ that we are optimizing.

The DPO objective effectively maximizes the likelihood of the model correctly ranking preferred responses higher than non-preferred ones, while maintaining reasonable proximity to the original reference model distribution. The sigmoid function $\sigma(\cdot)$ converts the log probability ratio differences into a probability space, allowing the model to learn from preference comparisons rather than absolute reward values.

## 4 Experiments

### 4.1 Dataset

We evaluate our method on two representative agent datasets that cover distinct interactive domains: WebShop (Yao et al., 2022) for web navigation and InterCode-SQL (Yang et al., 2023) for SQL database querying.

**WebShop**   WebShop presents a simulated e-commerce environment where agents must navigate through product listings, apply filters, and make purchase decisions based on specific user requirements. This dataset effectively captures the challenges of sequential decision-making in web interfaces, requiring agents to understand natural language instructions and translate them into appropriate navigation actions.

**InterCode-SQL**   InterCode-SQL focuses on database interaction scenarios where agents must formulate and execute SQL queries to retrieve, manipulate, or analyze data according to user specifications. This dataset tests the agent's ability to understand database schemas, compose syntactically correct SQL statements, and iteratively refine queries based on execution results.

Both WebShop and InterCode-SQL provide a dense reward scale from 0 to 1 to measure the task completion, enabling fine-grained evaluation of agent performance. We employ the average reward as the evaluation metric for all tasks, which effectively captures the overall performance across varying difficulty levels within each dataset.

## 4.2   Baseline

In our experimental evaluation, we compare our proposed Reward Rising Optimization (RRO) approach against several established baselines using Gemma-2$_{2B}$ Team et al. (2024) as the base model. These baselines represent the current state-of-the-art methods in agent planning for WebShop and InterCode-SQL tasks.

- **Few-shot:** This approach uses in-context learning with a small number of examples, without any additional training. It serves as our most basic baseline to demonstrate the performance gap between zero/few-shot prompting and more sophisticated techniques.
- **Supervised Fine-tuning (SFT) (Chen et al., 2023a):** Supervised Fine-Tuning uses direct imitation learning to train the model on expert demonstrations, capturing the mapping from inputs to desired outputs.
- **Exploration-based Trajectory Optimization (ETO) (Song et al., 2024):** Exploration-based Trajectory Optimization improves upon SFT by focusing on trajectories that maximize the outcome reward, showing modest improvements over standard SFT.
- **Iterative Step-level Process Refinement (IPR) (Xiong et al., 2024):** Iterative Step-level Process Refinement iteratively improves the agent's planning and execution steps through feedback on intermediate processes.
- **Fixed-sized Exploration:** We sample a fixed number of candidates for the next action and select the one with the highest reward as the chosen action and the one with the lowest reward as the rejected action. The key difference from our proposed RRO is the absence of the dynamic exploration of next actions and the reward rising selection criteria.

Our proposed RRO builds upon these process supervision approaches, introducing a novel optimization strategy that dynamically extends the exploration space. This approach achieves superior performance while significantly reducing computational costs, as demonstrated by the results in Table 1.

## 4.3   Implementation

We utilize the Gemma-2$_{2B\text{-base}}$ model as the foundation for our experiments, running on a computational infrastructure of 8 NVIDIA A100-80G GPUs. This hardware setup accommodates the demands of our preference optimization framework and supports extensive ablation studies across varying experimental configurations. During the supervised fine-tuning stage, we fine-tune the base model using expert trajectories from each dataset over 3 epochs, with a batch size of 48 and a learning rate of $2 \times 10^{-5}$. A cosine learning rate scheduler with a warmup ratio of 0.03 is employed to enhance training stability. In the

Table 1: Agent planning results of RRO and the baselines on Webshop and InterCode-SQL. The methods are using Gemma-2$_{2B}$ as the base model. (underline denotes the second-best performance; **bold** denotes the best performance; the improvement is measured against the second-best performing method.)

| Base Model | Method | Agent Planning | | | |
| --- | --- | --- | --- | --- | --- |
| | | WebShop | | InterCode-SQL | |
| | | Reward $\uparrow$ | # Sample $\downarrow$ | Reward $\uparrow$ | # Sample $\downarrow$ |
| Gemma-2$_{2B}$ | *Without Post-training* | | | | |
| | + Few Shot | 12.62 | 0 | 3.86 | 0 |
| | *Outcome Supervision* | | | | |
| | + SFT (Chen et al., 2023a) | 48.94 | 0 | 45.33 | 0 |
| | + ETO (Song et al., 2024) | 52.34 | 1 | 47.13 | 1 |
| | *Process Supervision* | | | | |
| | + IPR (Xiong et al., 2024) | 61.39 | 4 | 52.39 | 3 |
| | + Fixed-sized Exploration | 61.20 | 5 | 54.68 | 5 |
| | + **Reward Rising Optimization (RRO)** | **62.91** (+1.52) | 1.86 | **55.08** (+0.40) | 1.64 |

reward rising optimization sampling stage, candidate actions are sampled to identify those with a rising trend in process rewards. Sampling continues until an action yields a higher reward than the previous one or a maximum of 5 candidates is reached; if no rising reward is found, that step is omitted in the agent optimization phase. Each candidate action's process reward is estimated by averaging the outcome rewards of 5 generated rollouts, with a maximum of 10 rollout iterations. Finally, in the agent optimization stage, we apply the standard Direct Preference Optimization (DPO) algorithm to fine-tune the actor model over 3 epochs, using a batch size of 32, a constant learning rate of $3 \times 10^{-6}$, and a warmup ratio of 0.1. These hyperparameters are held consistent across all baseline and ablation settings.

## 4.4 Main Results

We compare our method with the strong baseline methods listed in Table 1. Based on the results, the proposed RRO demonstrates superior performance compared to other approaches across both WebShop and InterCode-SQL. RRO achieves the highest reward scores (62.91 for WebShop and 55.08 for InterCode-SQL) while requiring a reasonable number of samples (1.86 and 1.64 respectively). This represents a significant improvement over both outcome supervision methods (SFT and ETO) and other process supervision approaches (IPR and Fixed-sized Exploration). The results clearly show that RRO outperforms existing techniques, suggesting that the dynamic exploration strategy and reward rising criteria effectively balances exploration and reward maximization in agent planning scenarios. This empirical evidence strongly supports the effectiveness of the RRO method as a promising advancement in improving language model performance on complex reasoning tasks.

To further assess the robustness and scalability of Reward Rising Optimization (RRO), we extended our experiments to the larger Gemma-2$_{9B-base}$ model. On this model, SFT achieved reward scores of 63.59 on WebShop and 61.41 on InterCode-SQL; Fixed-sized Exploration improved performance to 68.87 and 65.30 respectively, using a fixed sample budget of 3. In contrast, RRO further elevated performance, reaching 71.24 on WebShop and 67.27 on InterCode-SQL, while requiring only 1.38 and 1.55 average samples respectively. These results confirm that RRO maintains its advantages even at larger model scales, achieving higher performance with improved sample efficiency. This successful scaling underscores the potential of RRO in complex, real-world environments.

## 4.5 Sampling Efficiency Analysis

Based on Figure 2, the results show that our proposed RRO significantly outperforms IPR in terms of sampling efficiency across both WebShop and InterCode-SQL tasks. For WebShop (Figure 2a), RRO achieves a high reward of 62.91 using only 1.86 sampled trajectories on average, while IPR requires 5 trajectories in the iterative DPO training to reach a comparable reward of 61.32, with performance improving gradually from 47.8 to 61.39 as trajectory

count increases. Similarly, for InterCode-SQL (Figure 2b), RRO attains a reward of 55.08 with just 1.64 sampled trajectories on average, whereas IPR shows a peak performance of 52.39 at 3 trajectories before declining to 42.5 at 5 trajectories. These results demonstrate RRO's superior efficiency, requiring fewer sampled trajectories to achieve higher rewards across both benchmarks.

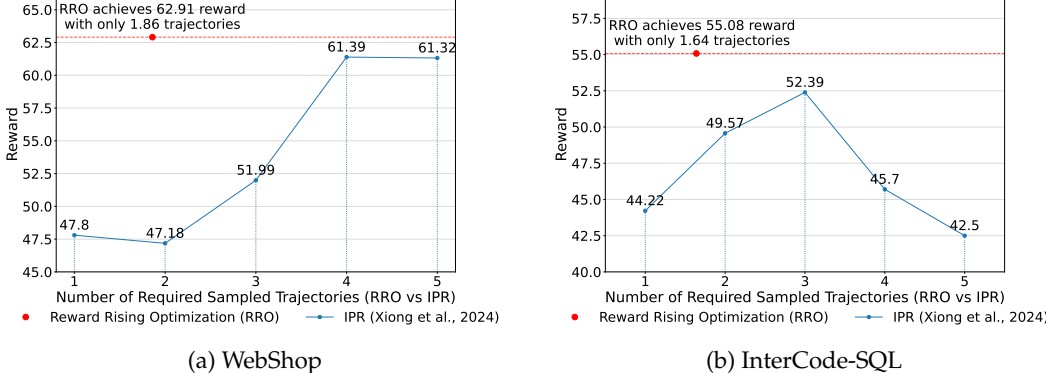

(a) WebShop          (b) InterCode-SQL

Figure 2: Sampling efficiency of RRO and IPR on WebShop and InterCode-SQL.

## 4.6 Rising Rewards Analysis

We analyze the reward tendency in the trajectories generated by the supervised fine-tuned model $\pi_{\text{SFT}}$ and the reward rising optimized model $\pi_{\text{RRO}}$. Specifically, we check if the neighboring actions demonstrate a rising tendency and calculate the proportion of actions with the rising rewards. We divide each trajectory into three equal stages (*Initial*, *Middle*, and *Final*), with each covering one-third of the trajectory length. As shown in Table 2, we find that $\pi_{\text{RRO}}$ consistently outperforms $\pi_{\text{SFT}}$ across both WebShop and InterCode-SQL datasets. On WebShop, our method shows modest improvements in the Middle (+1.20%) and Final (+1.36%) phases, while initially performing slightly lower (-0.58%). The benefits are more pronounced on InterCode-SQL, with significant gains across all stages: Initial (+6.75%), Middle (+2.08%), and Final (+4.42%). By the Final phase, our approach achieves 45.29% of actions with rising reward versus 40.87% for the baseline method, demonstrating more effective optimization and better reward alignment throughout the trajectory.

Table 2: Reward tendency of RRO and SFT on WebShop and InterCode-SQL in the Initial, Middle, and Final stages (each covering one-third of the trajectory length).

| Method | Proportion of Actions with Rising Reward (%) | | | | | |
| --- | --- | --- | --- | --- | --- | --- |
| | WebShop | | | InterCode-SQL | | |
| | Initial | Middle | Final | Initial | Middle | Final |
| Supervised Fine-tuning | **1.33** | 32.75 | 9.00 | 15.67 | 41.75 | 40.87 |
| Reward Rising Optimization | 0.75 (-0.58) | **33.95** (+1.20) | **10.36** (+1.36) | **22.42** (+6.75) | **43.83** (+2.08) | **45.29** (+4.42) |

## 5 Related Work

**Reinforcement Learning** Fine-tuning large language models (LLMs) with reinforcement learning has proven effective for aligning model outputs with user preferences. Instruct-GPT (Ouyang et al., 2022) demonstrated the potential of reinforcement learning from human feedback (RLHF) to improve truthfulness and reduce toxicity in generated outputs. However, RLHF's complexity and instability have motivated simpler alternatives, such as Direct Preference Optimization (DPO), which leverages a closed-form optimal policy to align LLMs with human preferences more efficiently (Rafailov et al., 2024). Proximal Policy Optimization (PPO) (Schulman et al., 2017), a foundational algorithm in reinforcement learning,

also underpins many RLHF methods by enabling stable policy updates through surrogate objectives. These advancements highlight the growing body of work aimed at enhancing the alignment and utility of LLMs.

**Process Reward Model**    Recent advancements in reinforcement learning have explored process reward models (PRMs) to enhance agent training by evaluating intermediate actions or decision-making steps. Christiano et al. (2017) introduced reward models based on human preferences, demonstrating the potential of fine-grained feedback for aligning agent behavior with human expectations. Bai et al. (2022) extended this by incorporating iterative feedback for multi-step tasks, enabling more effective supervision in reasoning-heavy domains. In the context of structured decision-making, Fu et al. (2022) proposed leveraging hierarchical rewards to guide agents through complex processes, improving sample efficiency and task generalization. These works collectively highlight the role of PRMs in bridging the gap between traditional RL and multi-step reasoning tasks.

## 6   Conclusion

In conclusion, we propose Reward Rising Optimization (RRO) which dynamically extends the next action exploration in training process reward models and focuses on the relative reward tendencies between reasoning steps. By identifying steps with rising rewards, the proposed RRO offers a more efficient and scalable solution to process reward modeling compared to traditional computational-intensive approaches. The mathematical foundations and empirical evidence across WebShop and InterCode-SQL underscore the potential of this technique to improve multi-step reasoning capabilities in large language models. Future work could explore the generalizability of this method across diverse task domains and further optimize the reward identification mechanism.

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
