# OpenReview forum: "RRO: LLM Agent Optimization Through Rising Reward Trajectories"
_colmweb.org/COLM/2025/Conference — COLM 2025_

### Official Review · Reviewer_QfKa · 2025-05-08

**Rating:** 6
**Confidence:** 3
**Ethics Flag:** 1

**Summary:**

This paper proposed Reward Rising Optimization (RRO), a dynamic exploration strategy for process supervision in LLM agent learning,aimed at improving process supervision in LLM agent learning.The approach prioritizes actions that exhibit a "rising reward" trend compared to preceding steps, offering an efficient strategy for exploring the action space. By dynamically adjusting the exploration scope, RRO helps in reducing computational costs while still capturing high-quality trajectory data. The method is empirically validated on WebShop and InterCode-SQL, showing significant improvements over existing benchmarks state-of-the-art approaches in terms of both performance and sample efficiency.

**Questions To Authors:**

please refer to section "Reasons To Reject"

**Reasons To Accept:**

1. The method provides a scalable solution for LLM-based agent training, offering clear advantages in real-world applications where computational resources are limited.The "rising reward" criterion balances exploration quality and computational cost, avoiding over- or under-exploration.
2. RRO is supported by robust theoretical explanations that show its potential for addressing critical challenges in multi-step agent tasks, reinforcing its applicability.
3. The experiments cover two distinct benchmarks, demonstrating RRO’s versatility and general applicability across different types of agents tasks.

**Reasons To Reject:**

1. The paper focuses primarily on controlled benchmarks (WebShop and InterCode-SQL), but it does not address how well the proposed method would generalize to real-world tasks or complex environments, which would provide a clearer picture of its practical utility.
2. The absence of an ablation study prevents a clear understanding of the contribution of key components in RRO.
3. The paper lacks analysis of average/max sampling steps required for convergence.
4. Does the baseline method (such as IPR) in this paper reproduce under the same hardware/hyperparameters? It is necessary to explain the consistency of the experimental Settings to avoid the conclusion being affected by the implementation differences.
5. Figure.2 lacks error bars or confidence intervals, obscuring the variance in reward scores.

---

> ### Author Response · Authors · 2025-06-02
>
> We thank **Reviewer QfKa** for their positive evaluation and for recognizing RRO's potential as a scalable solution, its balance of exploration and cost, its theoretical support, and its versatility across benchmarks.
>
> We address the reviewer’s concerns and questions as follows:
>
> ---
>
> > **Concern:** Generalization to real-world tasks or complex environments
>
> **Response:**
> Our choice of **WebShop** and **InterCode-SQL** was intentional, as both are established benchmarks that, while controlled, are specifically designed to capture key challenges reflective of real-world agent tasks.
>
> - **WebShop** simulates sequential decision-making in web navigation, requiring agents to comprehend natural language and translate user instructions into effective e-commerce interactions.
> - **InterCode-SQL** evaluates an agent’s ability to generate, execute, and iteratively refine SQL queries based on natural language instructions and execution feedback—closely mirroring real-world database usage.
>
> To further validate RRO’s robustness and scalability, we extended our experiments to the larger **Gemma-2-9B** model. The results are summarized below:
>
> #### Results on Gemma-2-9B:
>
> | Method                     | WebShop                    | InterCode-SQL              |
> |---------------------------|----------------------------|-----------------------------|
> | SFT                       | 0.6359                     | 61.41                       |
> | Fixed-sized Exploration   | 0.6887 (using 3 samples)   | 65.30 (using 3 samples)     |
> | Reward Rising Optimization| 0.7124 (avg. 1.38 samples) | 67.27 (avg. 1.55 samples)   |
>
> These findings show that **RRO continues to outperform baselines even at a larger model scale**, achieving higher performance with **notable sample efficiency** (1.38 and 1.55 average samples compared to the fixed budget of 3). This successful application on Gemma-2-9B underscores **RRO’s potential for scaling to more complex, real-world environments**. We will clarify our focus on representative benchmarks in the current work and outline future directions involving broader real-world tasks.
>
> ---
>
> > **Concern:** Absence of an ablation study
>
> **Response:**
> Our comparison against the **"Fixed-sized Exploration"** baseline in Table 1 serves as a core ablation. This baseline samples a fixed number of candidates without applying RRO's **dynamic "rising reward" stopping criterion**. RRO’s superior performance with fewer average samples directly highlights the benefit of its dynamic sampling strategy. We will make this ablation role more explicit in the text.
>
> ---
>
> > **Concern:** Lack of analysis of average/max sampling steps required for convergence
>
> **Response:**
> We report **average sample counts per actor step** in Table 1 (“#Sample”):  **1.86** for WebShop and **1.64** for InterCode-SQL
> using **Gemma-2-2B**. These values are substantially lower than those of fixed exploration baselines, demonstrating RRO’s sample efficiency.
>
> Additionally, **Figure 2** shows that RRO surpasses baseline methods **within a single DPO iteration**, indicating faster convergence.
>
> To clarify the sampling process: during training, actions are sampled **until one yields a higher reward than the previous**, or until a **maximum exploration size** is reached.  If no improvement is found within this limit, the current step is **skipped in downstream optimization**.  We will revise the text to clarify this mechanism and can also report observed sample ranges to provide a more comprehensive analysis.
>
> ---
>
> > **Concern:** Baseline method (e.g., IPR) reproducibility under the same hardware/hyperparameters & consistency of experimental settings
>
> **Response:**
> All experiments—including baselines—were run using **Gemma-2-2B** on the **same computational infrastructure**: 8 × **NVIDIA A100-80G GPUs**, ensuring fairness.
>
> For baseline implementations (e.g., IPR, ETO), we followed the descriptions provided in their original papers and reused their specified hyperparameters where applicable, adapting them appropriately to our base model. We will explicitly clarify the **consistency of the experimental setup** across all methods in the revised manuscript.
>
> ---
>
> We thank **Reviewer QfKa** again for their constructive comments, which will undoubtedly help improve the clarity and rigor of our paper.

---

> > ### Comment · Reviewer_QfKa · 2025-06-05
> >
> > I have read others' comments and the authors' response, I maintain my score.

---

### Official Review · Reviewer_a4Sq · 2025-05-10

**Rating:** 6
**Confidence:** 4
**Ethics Flag:** 1

**Summary:**

This paper proposes Reward Rising Optimization (RRO), a new approach to process supervision in LLM-based agent learning. RRO focuses on relative reward increases --- sampling is stopped once an action yields a higher process reward than the previous step. This strategy improves sample efficiency and reduces computation cost. Authors demonstrate that RRO is validated with mathematical justifications and evaluated on WebShop and InterCode-SQL, and show that it outperforms prior methods like IPR and ETO.

**Questions To Authors:**

- How does RRO behave in environments where rewards are noisy or non-monotonic across good trajectories?
- Have you explored using a threshold delta (e.g., reward increase ≥ 0.05) instead of any positive increase?
- Can this method be applied to domains without dense reward functions or where the reward model is unreliable?

**Reasons To Accept:**

- Introduces a conceptually simple yet effective sampling stopping criterion based on reward increase.
- Demonstrated ability to achieve comparable or better performance with significantly fewer samples.
- The method is well-motivated, and Algorithm 1 is easy to understand and replicable.

**Reasons To Reject:**

- Simple experiments. The authors only conduct experiments on a small model Gemma-2B on two tasks. I would expect some experiments on at least 7B models to show that the method is still effect.
- No justification of fewer samples. Although the experiments show that the RRO methods achieves the similar results as IPR, and other methods while with fewer samples been explored. There is no proof of it, I would concern in some cases, the RRO will need more exploration compared to other methods. The math ground of the existence does not mean it is easier to find.
- What happens when rewards are flat or noisy? Is the “first rising” heuristic always reliable?

---

> ### Author Response · Authors · 2025-06-02
>
> We thank **Reviewer a4Sq** for their constructive feedback and for recognizing the conceptual simplicity and effectiveness of RRO's sampling criterion, its ability to achieve strong performance with fewer samples, and the clarity of Algorithm 1.
>
> We address the reviewer’s concerns as follows:
>
> ---
>
> > **Concern:** Simple experiments. The authors only conduct experiments on a small model Gemma-2B on two tasks. I would expect some experiments on at least 7B models to show that the method is still effective.
>
> **Response:**
> We appreciate the reviewer’s request for validation on larger models. Our initial choice of **Gemma-2-2B** and the **WebShop** and **InterCode-SQL** benchmarks was intended to clearly demonstrate RRO’s core advantages on established tasks.
>
> To further support the method’s scalability, we tested RRO on Gemma-2-9B. This extended evaluation compared RRO against SFT and Fixed-sized Exploration, with exploration/sample sizes capped at 3 due to rebuttal time constraints (more comprehensive runs are planned for the final manuscript).
>
> #### Results on Gemma-2-9B:
>
> | Method                     | WebShop                    | InterCode-SQL              |
> |---------------------------|----------------------------|-----------------------------|
> | SFT                       | 0.6359                     | 61.41                       |
> | Fixed-sized Exploration   | 0.6887 (using 3 samples)   | 65.30 (using 3 samples)     |
> | Reward Rising Optimization| 0.7124 (avg. 1.38 samples) | 67.27 (avg. 1.55 samples)   |
>
> The Gemma-2-9B results are highly promising. RRO maintains a clear performance advantage over both SFT and Fixed-sized Exploration. Critically, this superior performance is achieved with greater sample efficiency—RRO averaged 1.38 and 1.55 samples on WebShop and InterCode-SQL respectively, well below the baseline's fixed budget of 3. These findings bolster our confidence in RRO's scalability and its value for enhancing larger language models.
>
> ---
>
> > **Concern:** No justification of fewer samples. Although the experiments show that the RRO methods achieve similar results as IPR and other methods while exploring fewer samples, there is no proof of it. I would be concerned in some cases the RRO will need more exploration compared to other methods. The math ground of the existence does not mean it is easier to find.
>
> **Response:**
> This is an insightful point. Our mathematical grounding (Section 3.3) proves the existence of an action​ with a process reward greater than or equal to the preceding action's reward, but this doesn't guarantee it's found on the first try. In our RRO implementation, there's a **practical trade-off between the quality of the sampled action and the computational cost**. Specifically, sampling continues until an action's reward is higher than the previous one or a maximum exploration size is reached. If no action with a rising reward is found within this limit, that step is skipped in the subsequent agent optimization.
>
> We typically set **RRO's maximum exploration limit** to be the same as the number of samples used by **Fixed-sized Exploration**. This means that even in the **worst-case scenario** (where RRO explores up to its maximum limit), its sampling count will not exceed that of the fixed baseline for that step.
>
> **Empirically**, as shown in Table 1 (e.g., #Sample for RRO is 1.86 for WebShop with Gemma-2-2B) and Figure 2, RRO consistently requires, on average, significantly fewer samples than fixed-exploration strategies. This strongly suggests that, in practice, actions satisfying the "rising reward" criterion are found relatively quickly within these benchmarks.
>
> ---

---

> > ### Author Response · Authors · 2025-06-02
> >
> > > **Concern:** What happens when rewards are flat or noisy? Is the “first rising” heuristic always reliable?
> >
> > **Response:**
> > This is a crucial question regarding the robustness of the "first rising" heuristic. In environments with perfectly flat or highly noisy reward landscapes, the quality of preference data generated by the current heuristic might diminish.
> >
> > As you suggested, **incorporating a threshold delta** (e.g., requiring the reward to rise by at least $\delta > 0$) is a promising direction. This could enhance the quality of curated data by ensuring a more significant rising tendency. However, it might also **increase the average number of samples needed** to find such a candidate, potentially impacting overall efficiency.
> >
> > While RRO demonstrates superior efficiency in its current form, exploring a threshold delta is a **valuable avenue for future experiments**, and we plan to investigate this.
> >
> > ---
> >
> > > **Concern:** How does RRO behave in environments where rewards are noisy or non-monotonic across good trajectories?
> >
> > **Response:**
> > As discussed above, **high noise could affect the current stopping condition**. If optimal trajectories inherently involve temporary, strategic dips in reward (i.e., non-monotonic but globally effective paths), the current greedy "first rising" approach might prematurely abandon such lines of exploration.
> >
> > Empirically, for the tasks in our chosen benchmarks (WebShop and InterCode-SQL), this strategy has led to **improved performance with enhanced efficiency**. In environments characterized by significant noise or non-monotonic reward trajectories, we intuitively consider **applying the rising reward optimization more selectively**—for example, emphasizing it more in the later stages of a problem-solving trajectory, where a consistent upward trend in rewards might be more critical, as suggested by our findings in Table 2.
> >
> > ---
> >
> > > **Concern:** Can this method be applied to domains without dense reward functions or where the reward model is unreliable?
> >
> > **Response:**
> > RRO, in its current formulation, **relies on a meaningful process reward model ($r_\text{PRM}$)**. If the underlying outcome reward model ($r_\text{ORM}$) is very sparse, or if the resulting $r_\text{PRM}$ is unreliable, then the guidance RRO provides will naturally be compromised.
> >
> > This is a **challenge shared by most reinforcement learning algorithms** that depend on the quality of the reward signal. Therefore, RRO's applicability is intrinsically linked to the **ability to learn or define a sufficiently informative process reward model** for the given domain.
> >
> > ---
> >
> > We thank you again for these **thought-provoking questions** and **valuable suggestions**.

---

> > > ### Comment · Reviewer_a4Sq · 2025-06-05
> > >
> > > These explanation has addressed most of my concerns. I will raise my score.

---

### Official Review · Reviewer_w85m · 2025-05-11

**Rating:** 6
**Confidence:** 4
**Ethics Flag:** 1

**Summary:**

This paper addresses the challenge of training LLM-based agents to perform complex multi-step reasoning tasks. Such tasks often fail due to their sensitivity to key decision points and the difficulty of correcting subtle trajectory errors. While Process Reward Models (PRMs) have been proposed to provide fine-grained feedback for each step, their scalability is hindered by the high computational cost of exploring every possible action at each step.

To mitigate this, the authors propose Reward Rising Optimization (RRO), a new method that emphasizes the relative reward trend across sequential steps. RRO dynamically scales the sampling process and halts exploration once an action yields a higher process reward than its predecessor. This strategy prioritizes high-reward trajectories while reducing computational overhead. The paper provides both theoretical justifications and empirical evidence from experiments on the WebShop and InterCode-SQL benchmarks.

**Questions To Authors:**

To my understanding, RRO halts sampling as soon as it observes a rising reward. What if a better (higher-reward) action exists further in the sampling sequence? Does this mean RRO is not guaranteed to find the globally optimal trajectory?

Comments:
The Gemma-2-2B model is used in all experiments but is not properly cited in the main text. Please include a citation for clarity and reproducibility.

**Reasons To Accept:**

1. **Novel Approach for Agent Training**: The paper introduces RRO that focuses on the relative reward tendency and dynamic adjustment of the search space. While it bears resemblance to classical search heuristics, its application to process supervision in agent learning offers an effective alternative to exisiting PRM schemes.

2. **Improved Empirical Results and Sampling Efficiency**: The method achieves state-of-the-art performance on WebShop and InterCode-SQL, outperforming both outcome-supervised baselines (SFT, ETO) and process-supervised baselines (IPR, fixed-size exploration). Notably, it achieves higher rewards while requiring fewer sampled trajectories, demonstrating better sampling efficiency.

3. **Theoretical Grounding**: The authors provide theoretical justification for the reward rising criterion, showing that under the Bellman equation and process reward formulation, there always exists a future action with equal or higher expected reward. This adds credibility to the method's stopping criterion.

**Reasons To Reject:**

1. **Limited Generalization Across Tasks and Models**: The method is evaluated on only two agent benchmarks, which—although representative—are limited in diversity. Furthermore, experiments are conducted using only a single model (Gemma-2-2B). It remains unclear how RRO would generalize to models of different architectures or other domains.

2. **Implementation Complexity and Opaque Computational Trade-offs**: The overall pipeline includes supervised fine-tuning, reward rising sampling, and DPO training. While modular, each stage introduces its own complexity. Reproducing the full pipeline may require substantial engineering effort, particularly in large-scale environments. Although RRO is shown to be more sampling-efficient, the total computational cost (e.g., GPU hours) is not reported. Since the method still relies on Monte Carlo tree search and dynamic sampling, a more comprehensive comparison of computational cost would help practitioners assess its practicality.

---

> ### Author Response · Authors · 2025-06-02
>
> We thank **Reviewer w85m** for their thorough review and insightful comments. We are pleased the reviewer recognized the novelty of RRO's approach to agent training, its improved empirical results and sampling efficiency, and the value of its theoretical grounding.
>
> We address the reviewer’s concerns as follows:
>
> ---
>
> > **Concern:** Limited Generalization Across Tasks and Models
>
> **Response:**
> We appreciate the reviewer highlighting the importance of evaluating RRO's generalization capabilities. Our initial selection of **WebShop** and **InterCode-SQL** aimed to test RRO on established and distinct agent task domains. Similarly, our primary experiments utilized the **Gemma-2-2B** model to rigorously demonstrate the foundational principles and efficiencies of RRO.
>
> To further address the reviewer's valid point on model scalability and broaden our empirical validation, we have now extended our evaluation to the **larger Gemma-2-9B** model. We benchmarked RRO against **SFT** and **Fixed-sized Exploration** on this larger model. For these new experiments, the exploration limit for RRO and the fixed size for the "Fixed-sized Exploration" baseline were set to 3, primarily due to time constraints for this rebuttal phase. We intend to conduct more exhaustive runs, potentially with varied exploration settings, for our final manuscript.
>
> #### Results on Gemma-2-9B:
>
> | Method                  | WebShop (score)        | InterCode-SQL (score)  |
> |------------------------|------------------------|-------------------------|
> | SFT                    | 0.6359                 | 61.41                   |
> | Fixed-sized Exploration (using 3 samples) | 0.6887 | 65.30                   |
> | Reward Rising Optimization (avg. 1.38 / 1.55 samples) | **0.7124** | **67.27**           |
>
> The results from the **Gemma-2-9B** model are very encouraging. They show that **RRO not only maintains its performance advantage over SFT and Fixed-sized Exploration** but does so with remarkable sample efficiency. For instance, RRO achieved superior scores on both WebShop and InterCode-SQL while exploring, on average, significantly fewer candidates (1.38 and 1.55, respectively) than the Fixed-sized Exploration's budget of 3. This successful application to a larger model reinforces our confidence in **RRO's scalability** and its potential to enhance agent learning efficiency across different model sizes.
>
> ---
>
> > **Concern:** Implementation Complexity and Opaque Computational Trade-offs
>
> **Response:**
> We appreciate the reviewer's point regarding implementation complexity and computational trade-offs. Our full pipeline comprises **Supervised Fine-tuning (SFT)**, our novel **Reward Rising Optimization (RRO)** sampling stage, and **Direct Preference Optimization (DPO)**. For the SFT and DPO stages, we leveraged standard implementations, utilizing established frameworks like FastChat [1] and TRL [2], ensuring these components align with common practices.
>
> The primary contribution of RRO, and where its efficiency gains originate, is the **dynamic "reward rising" sampling strategy** for collecting preference data. This strategy is specifically designed to reduce the number of the sampled action candidates and avoid over- or under-exploration.
>
> It's important to clarify that **RRO lies in the sampling strategy** (i.e., when to stop exploring candidates for a given step), while the fundamentally low-level implementation of how individual candidates are sampled or their rewards estimated is the same. Therefore, the **average number of samples would be sufficient** to demonstrate the higher efficiency of RRO.
>
> Specifically, many baselines involving **MCTS-based process reward estimation** typically sample a fixed number of K candidates, estimate the process reward for all K of them, and then select the preference pair. In contrast, **our RRO strategy estimates the process reward for each one right after the action is sampled**. Exploration for the current step terminates as soon as an action satisfying our "rising reward" criterion is identified. This dynamic strategy makes it possible to sample fewer actions than the baselines. Moreover, we also **set a maximum exploration size for RRO**, which is set to be the same as the K used in the "Fixed-sized Exploration" baseline. This means RRO always explores equal or fewer candidates than the Fixed-sized Exploration.
>
> As empirically demonstrated and reported in **Table 1**, RRO typically requires significantly fewer samples on average (1.86 for WebShop and 1.64 for InterCode-SQL on Gemma-2-2B) to find a satisfactory action. This demonstrates substantially **higher average-case efficiency** in terms of the number of action candidates evaluated, and thus reduced overall computational cost for data collection, compared to baselines that rely on fixed-size MCTS estimations for every step.
>
> ---

---

> > ### Author Response · Authors · 2025-06-02
> >
> > > **Concern:** To my understanding, RRO halts sampling as soon as it observes a rising reward. What if a better (higher-reward) action exists further in the sampling sequence? Does this mean RRO is not guaranteed to find the globally optimal trajectory?
> >
> > **Response:**
> > This is an excellent question. The stopping criterion in RRO is a **heuristic** designed to strike a balance between exploration cost and reward quality. While it does **not guarantee the discovery of the globally optimal action** at each step, it reliably identifies actions that reflect an improvement relative to the previous step. This is grounded in our theoretical grounding, which ensures the existence of at least one such rising-reward action under reasonable assumptions.
> >
> > The key insight behind RRO is that **consistently identifying "rising reward" steps enables the agent to construct high-quality trajectories more efficiently than exhaustive or fixed-size exploration**. Our empirical results (Table 1), where RRO surpasses baselines with more extensive sampling, validate the effectiveness of this trade-off in practice.
> >
> > It is important to note that **exhaustively evaluating all possible actions is infeasible** in practice due to the vastness of the action space in our benchmarks (e.g., the space of all possible SQL queries in InterCode-SQL). Therefore, global optimality is rarely attainable for any method operating under practical resource constraints.
> >
> >
> > ---
> >
> > > **Comment:** The Gemma-2-2B model is used in all experiments but is not properly cited in the main text. Please include a citation for clarity and reproducibility.
> >
> > **Response:**
> > Thank you for pointing this out. We will ensure that **Gemma-2-2B is properly cited** where it's first mentioned (e.g., Sections 4.2 and 4.3) in the camera-ready version.
> >
> > ---
> >
> > We appreciate the reviewer's thoughtful engagement and believe these clarifications and planned revisions will strengthen the paper.
> >
> > ---
> >
> > **References**
> > [1] FastChat: https://github.com/lm-sys/FastChat
> > [2] TRL: https://github.com/huggingface/trl

---

> > > ### Comment · Reviewer_w85m · 2025-06-06
> > >
> > > Thank you for the thoughtful response. I have made necessary adjustment of my evaluation based on it.

---

### Official Review · Reviewer_yiKS · 2025-05-12

**Rating:** 7
**Confidence:** 5
**Ethics Flag:** 1

**Summary:**

This paper focuses on the efficiency problem on the process supervision for LLM agent learning. To effectively acquire the training data throug the per-step trajectory exploration, this paper propose Reward Rising Optimization method. This method dynamically expands the search space for the next action candidates. Experiments show the effectiveness of the method.

**Reasons To Accept:**

* The proposed RRO method introduces a dynamic search space expansion mechanism for process reward-based LLM agent training, offering a fresh perspective on efficiency optimization.
* Experiments show that RRO outperforms both outcome-based and process-based reward methods while requiring fewer exploration samples.
* This paper is well-written.

**Reasons To Reject:**

* The experiments lack detailed setup. Key hyperparameters and implementation specifics are missing, making reproducibility difficult.
* Experiments are conducted only on Gemma-2-2B. Broader validation on larger models (e.g., 7B or larger LLMs) would strengthen the claims.

---

> ### Author Response · Authors · 2025-06-02
>
> We sincerely thank **Reviewer yiKS** for their positive assessment and constructive feedback. We are encouraged that the reviewer found our proposed **RRO** method novel, the efficiency optimization a fresh perspective, the experimental results strong, and the paper well-written.
>
> We address the reviewer’s concerns as follows:
>
> ---
>
> > **Concern:** Lack of Experimental Details
>
> **Response:**
> We appreciate the reviewer pointing out the need for a more detailed experimental setup. We are providing the key hyperparameters and implementation specifics below, and these will be included in a dedicated Appendix in the revised manuscript to enhance reproducibility.
>
> - **Supervised Fine-tuning (SFT) Stage:**
>   We utilized expert trajectories provided by each dataset to fine-tune the base model (**Gemma-2-2B**). The training was configured with 3 epochs, a batch size of 48, and a learning rate of 2e-5. To ensure training stability, we employed a cosine learning rate scheduler with a warmup ratio of 0.03.
>
> - **Reward Rising Optimization (RRO) Sampling Stage:**
>   As described in the paper, we sample next action candidates aiming to identify actions with process rewards demonstrating a rising trend. Specifically, sampling continues until an action's reward is higher than the previous one or a **maximum exploration size of 5** is reached. If no action with a rising reward is found within this limit, that step is skipped in the subsequent agent optimization phase. The **process reward** for each candidate action is estimated by averaging the **outcome rewards of 5 generated rollouts**. The **maximum iteration rounds** for rollouts was set to 10.
>
> - **Agent Optimization Stage:**
>   We followed the standard **Direct Preference Optimization (DPO)** algorithm. The actor model was optimized over 3 epochs with a batch size of 32, a constant learning rate of 3e-6, and a warmup ratio of 0.1. These hyperparameters were consistently applied across all baseline comparisons and ablation studies.
>
> ---
>
> > **Concern:** Experiments Conducted Only on Gemma-2-2B
>
> **Response:**
> This is a valid point. Our initial experiments focused on Gemma-2-2B to establish the core efficacy of RRO due to resource considerations. We believe the significant improvements shown on this model class (as seen in Table 1 and Figure 2) provide a strong proof-of-concept for RRO's benefits.
>
> We agree that validating RRO on larger models would indeed strengthen our claims about broader applicability. To address this, we conducted **additional experiments with a larger base model, Gemma-2-9B**, comparing RRO against SFT and Fixed-sized Exploration.  Due to time constraints, we reduced the maximum exploration size for RRO and the fixed size for Fixed-sized Exploration from 5 to 3 to curate preference data more rapidly; this might have slightly impacted the final performance. We plan to conduct a more comprehensive comparison, including runs with the original exploration sizes, for the final version.
>
> #### Results on Gemma-2-9B:
>
> | Method                 | WebShop (score) | InterCode-SQL (score) |
> |------------------------|------------------|------------------------|
> | SFT                    | 0.6359           | 61.41                  |
> | Fixed-sized Exploration (size=3) | 0.6887           | 65.30                  |
> | Reward Rising Optimization (avg size: 1.38 / 1.55) | **0.7124** | **67.27**               |
>
> We find that even with a larger model like **Gemma-2-9B**, **RRO continues to outperform both SFT and Fixed-sized Exploration** on both WebShop and InterCode-SQL benchmarks. Notably, RRO achieves these superior results while maintaining a **lower average exploration size** (1.38 for WebShop and 1.55 for InterCode-SQL) compared to the Fixed-sized Exploration baseline (fixed size=3). This further underscores the **efficiency and effectiveness** of the RRO strategy, suggesting its potential benefits extend to larger language models.
>
> ---
>
> We thank **Reviewer yiKS** again for their valuable feedback, which will help us improve the paper.

---

> > ### Comment · Reviewer_yiKS · 2025-06-05
> >
> > This explanation has addressed most of my concerns. Additional experiments on a larger model (Gemma-2-9B) further demonstrate the effectiveness of the proposed method. Overall, this is a good paper on LLM agents, and I maintain my ​​"accept"​​ score.

---

### Decision · Program_Chairs · 2025-07-08

**Decision:**

Accept

**Comment:**

This work proposes a search heuristic that focuses on promising trajectories for better PRMs.
All reviewers agree that this sampling strategy is a nice contribution that leads to better performance and efficiency. The authors also added experiments with a second model during the author response.